# Proteomics of Duchenne Muscular Dystrophy Patient iPSC-Derived Skeletal Muscle Cells Reveal Differential Expression of Cytoskeletal and Extracellular Matrix Proteins

**DOI:** 10.3390/cells14211688

**Published:** 2025-10-28

**Authors:** Sarah-Marie Gallert, Mitja Fölsch, Lampros Mavrommatis, Urs Kindler, Karin Schork, Martin Eisenacher, Matthias Vorgerd, Beate Brand-Saberi, Britta Eggers, Katrin Marcus, Holm Zaehres

**Affiliations:** 1Department of Anatomy and Molecular Embryology, Institute of Anatomy, Faculty of Medicine, Ruhr University Bochum, 44801 Bochum, NRW, Germany; sarah-marie.gallert@rub.de (S.-M.G.); mitja.foelsch@rub.de (M.F.); lampros.mavrommatis@rub.de (L.M.); urs.kindler@rub.de (U.K.); beate.brand-saberi@rub.de (B.B.-S.); 2Medical Proteome Center, Faculty of Medicine, Ruhr University Bochum, 44801 Bochum, NRW, Germany; karin.schork@rub.de (K.S.); martin.eisenacher@rub.de (M.E.); britta.eggers@rub.de (B.E.); katrin.marcus@rub.de (K.M.); 3Medical Proteome Analysis, Center for Proteindiagnostics (PRODI), Faculty of Medicine, Ruhr University Bochum, 44801 Bochum, NRW, Germany; 4Department of Neurology with Heimer Institute for Muscle Research, Faculty of Medicine, University Hospital Bergmannsheil, Ruhr University Bochum, 44789 Bochum, NRW, Germany; matthias.vorgerd@bergmannsheil.de; 5Core Unit for Bioinformatics (CUBiMed.RUB), Medical Faculty, Ruhr University Bochum, 44801 Bochum, NRW, Germany

**Keywords:** Duchenne Muscular Dystrophy, patient-induced pluripotent stem cells, proteomics, skeletal muscle, organoids, disease modelling, mass spectrometry, biomarker

## Abstract

Proteomics of dystrophic muscle samples is limited by the amount of protein that can be extracted from patient biopsies. Cells and tissues derived from patient-derived induced pluripotent stem cells (iPSCs) can be an expandable alternative source. We have patterned iPSCs from three Duchenne muscular dystrophy (DMD) patient lines into skeletal muscle cells using a two-dimensional as well as our three-dimensional organoid differentiation system. Probes with sufficient protein amounts could be extracted and prepared for mass spectrometry. In total, 3007 proteins in 2D and 2709 proteins in 3D were detected in DMD patient probes. A total of 83 proteins in 2D and 338 proteins in 3D can be described as differentially expressed between DMD and control patient probes in a post hoc test. We have identified and we propose Myosin-9, Collagen 18A, Tropomyosin 1, BASP1, RUVBL1, and NCAM1 as proteins specifically altered in their expression in DMD for further investigation. Proteomics of skeletal muscle organoids resulted in greater consistency of results between cell lines in comparison to the two-dimensional myogenic differentiation protocol.

## 1. Introduction

Duchenne muscular dystrophy (DMD) is an X-linked neuromuscular disease with an incidence in 1 in 3500 to 5000 newborn boys caused by mutations in the Dystrophin (DMD) gene, the largest gene known to humans [1]. Dystrophin has been shown to stabilize the sarcolemma against contraction-induced mechanical stress by linking intracellular components of the cytoskeleton to the Dystrophin–glycoprotein complex (DGC), a transmembrane complex which is in turn connected to the extracellular matrix (ECM) [2]. Deletions are the most common type of mutation causing DMD; among these, out-of-frame mutations are often associated with the milder Becker Muscular Dystrophy (BMD) phenotype [3]. In DMD patients, the loss of Dystrophin function leads to a reduction in myofiber stability, resulting in necrosis of skeletal and cardiac muscle cells [4]. The associated inflammatory reaction, in turn, leads to fibrotic remodeling and, finally, fatty degeneration of the muscle tissue [5]. DMD typically manifests in boys between the ages of three and five years in the form of progressive skeletal muscle weakness. Loss of ambulatory ability is reported between the ages of nine and thirteen years. Spanning about two to three decades, the life expectancy of patients is relatively short; the main complications arise from pulmonary and cardiac failure [4]. BMD patients usually exhibit a later onset as well as a milder course of the disease, with patients losing ambulation between the ages of 16 and 80 years [4]. However, it has been shown that DMD does not only affect myofiber stability, but also that muscle regeneration from satellite cells is also impaired in DMD patients [6]. Furthermore, DMD patients exhibit a central nervous system component of the disease, as several dystrophin isoforms are also expressed in the brain [7]. This often includes a reduction in cognitive abilities but can also include neuropsychiatric comorbid conditions such as autism spectrum disorders [8].

As DMD is not simply an issue with myofiber stability, it is important to examine not only single proteins but also differentially expressed proteins on a larger scale as well as pathways to identify potential therapeutic candidates and biomarkers. In this study, we aim to analyze the proteome of muscle cells cultured from patient-derived induced pluripotent stem cells (iPSCs). Previously, studies on this topic have primarily focused on the mdx mouse model of Duchenne muscular dystrophy and on identifying human serum biomarkers of DMD, although a few studies have also examined the proteomes of muscle biopsies from DMD patients [9,10,11,12,13].

Since their development in 2007, iPSCs have enabled the study of diseases in cell culture models without requiring patient biopsies or embryonic stem cells, as iPSCs can be generated from adult fibroblasts [14]. In the present study, we differentiated iPSCs from DMD patients into skeletal muscle cells using both an established 2D differentiation protocol and our protocol for generating three-dimensional skeletal muscle organoids [15,16,17]. Limitations of 2D differentiation, including the lack of structural remodeling during myogenesis and the incomplete maturation of Pax7^+^ myogenic stem/progenitor pools in vitro, have led us to the development of the organoid approach, which recapitulates differentiation steps like paraxial mesoderm, neuromesodermal progenitors, and dermomytome toward fetal myogenesis. Specifically, myogenic stem/progenitor populations in their dormant, activated, and committed states can be expanded in the organoid system [18]. Methodologically, our previous description of this system provided a comprehensive transcriptomic evaluation with RNA-seq and scRNA-seq data sets of WT and DMD lines [18,19]. In the current study, we are providing novel proteome data sets for the 2D as well as our 3D system, thereby also further validating the organoid approach through DMD-specific proteomics.

## 2. Materials and Methods

### 2.1. Human Induced Pluripotent Stem Cell (iPSC) Cultures

Duchenne muscle dystrophy patient iPSC lines:

“DMD1” (DMD_iPS1) (passage 30–50) [20]

“DMD2” (DMD_iPS2) (passage 30–50) [20]

“DMD3” (iPSCORE_65_1, UCSD061i-65-1, WiCell #WB60393) (passages 20–50) [21]

Wild Type iPSC line:

“WT” Cord Blood iPSC (CB CD34+ iPSC) (passages 23–50) [22].

Human induced pluripotent stem cell lines, DMD1, DMD2, DMD3 and WT iPSC were cultured in TESR-E8 (StemCell Technologies, Vancouver, Canada) on Matrigel GFR (Corning, Corning, NY, USA)-coated 6-well plates and 12-well plates [20,21,22].

### 2.2. Skeletal Muscle Differentiation

Skeletal muscle differentiation in the two-dimensional culture system was performed according to a published protocol [15,23]. Based on mimicking the normal skeletal muscle development stages without the usage of transgenes, the protocol follows a sequence of different media with small molecules. Feeder-free iPSCs were cultured with a density of 10–20%. The induction of skeletal muscle differentiation was conducted based on serum-free conditions. In the first 6 days of skeletal muscle differentiation, 3 μM CHIR-99021 (SigmaAldrich, St. Louis, MO, USA), 0.5μM LDN-193189 (SigmaAldrich, St. Louis, MO, USA) and 10 ng/μL FGF-2 (ThermoFisher Scientific, Waltham, MA, USA) were used as supplements. Beginning on the 6th day of differentiation, Knock-out serum replacement (KSR) as a media supplement was introduced with a final concentration of 15%. IGF-1 at 2 ng/μL (ThermoFisher Scientific, Waltham, MA, USA) and HGF at 10 ng/μL (ThermoFisher Scientific, Waltham, MA, USA) are especially important growth factors for the myogenic differentiation. Media change took place daily in the first 12 days of differentiation, followed later by media change every other day. By following this protocol, spontaneously twitching myofibers could be seen around day 25.

Three-dimensional skeletal muscle organoid differentiation was induced as follows [16,17,18]: Prior to differentiation, undifferentiated human PSCs, 60–70% confluent, were enzymatically detached and dissociated into single cells using TrypLE Select (ThermoFisher Scientific, Waltham, MA, USA). Embryoid bodies formed via the hanging drop approach, with each droplet containing 3–4 × 10^3^ human single PSCs in 20 μL, were cultured hanging on TESR-E8 supplemented with Polyvinyl Alcohol (PVA) at 4 mg/mL (Sigma Aldrich, St. Louis, MO, USA) and rock inhibitor (Y-27632) at 10 μM (StemCell Technologies, Vancouver, BC, Canada) on the lid of Petri dishes. At the beginning of skeletal muscle organoid differentiation, embryoid bodies the size of 250–300 μm were embedded into Matrigel and cultured in DMEM/F12 basal media (ThermoFisher Scientific, Waltham, MA, USA) supplemented with Glutamine (ThermoFisher Scientific, Waltham, MA, USA), Non-Essential Amino Acids (ThermoFisher Scientific, Waltham, MA, USA), 100× ITS-G (ThermoFisher Scientific, Waltham, MA, USA), 3 μM CHIR99021 (SigmaAldrich, St. Louis, MO, USA), and 0.5 μM LDN193189 (SigmaAldrich, St. Louis, MO, USA). On Day 3, human recombinant basic Fibroblast Growth Factor (bFGF) (ThermoFisher Scientific, Waltham, MA, USA) at a final concentration of 10 ng/μL was added to the media. Subsequently, on Day 5, the concentration of bFGF was reduced to 5 ng/μL and the media were further supplemented with 10 nM Retinoic Acid (SigmaAldrich, St. Louis, MO, USA). The differentiation media on Day 7 was supplemented only with human recombinant Sonic hedgehog (hShh) (ThermoFisher Scientific, Waltham, MA, USA) at 34 ng/μL, human recombinant WNT1A (ThermoFisher Scientific, Waltham, MA, USA) at 20 ng/μL, and 0.5 μM LDN193189. On Day 11, the cytokine composition of the media was changed to 10 ng/μL of bFGF and 10 ng/μL of human recombinant Hepatocyte Growth Factor (HGF) (ThermoFisher Scientific, Waltham, MA, USA). From Day 15 onwards, the basal media were supplemented with ITS-X (ThermoFisher Scientific, Waltham, MA, USA) and human recombinant HGF at 10 ng/μL. During the first 3 days of the differentiation, the media were changed daily, every second day from the 3rd till the 30th day of differentiation, and every third day from Day 30 onwards.

### 2.3. Sample Preparation for Mass Spectrometry

Extracts were taken from the 2D cultures and the 3D cultures on Day 35. Samples were briefly centrifuged to remove culture fluid or PBS. To each sample, 50 µL of DIGE buffer was added. Each sample was homogenized with a pestle and then sonicated for 6 rounds of 10 s each. The supernatant was collected for further analysis.

For the tryptic digestion, an in-gel-digestion approach was applied. To the protein solution equivalent to 50 µg of Protein, 4× LDS Buffer and 2 M dithiothreitol (DTT) were added. Samples were incubated at 95 °C for 10 min and loaded onto a 12% BisTris-Gel. Gel electrophoresis was performed out for 15 min at 50 V and 2 min at 180 V, respectively, in MOPS buffer, after which the gels were stained with Coomassie solution. Protein bands were excised, destained with ammonium bicarbonate/acetonitrile, and digested overnight at 37 °C with trypsin (1:40, *w*/*w*, in 50 mM ammonium bicarbonate). Digestion was stopped by adding 50 µL of a 1:1 (*v*/*v*) 0.1% trifluoroacetic acid (TFA)/acetonitrile solution to the samples. After incubation in an ultrasonic bath for 15 min to elute the digested proteins, the solution was aspirated, transferred to a new sample vial and dried in a vacuum centrifuge. Peptides were resuspended in 30 µL 0.1% TFA and stored at −80 °C until further use.

### 2.4. Proteomic Analysis Using Mass Spectrometry

The peptide concentrations were determined via amino acid analysis as previously described [24]. Liquid chromatography tandem mass spectrometry (LC-MS/MS) analyses were performed with 200 ng of the sample per run using an UltiMate 3000 RSLC nano LC system (ThermoFisher Scientific, Bremen, Germany) coupled to an Orbitrap Elite mass spectrometer (ThermoFisher Scientific, Bremen, Germany). Moreover, equal amounts of each sample were taken to prepare a mixture, which was measured in between the samples, allowing the analysis of technical variances during measurements.

The peptides were first loaded onto a capillary pre-column (ThermoFisher Scientific, Bremen, Germany, 100 μm × 2 cm, particle size 5 μm, pore size 100 Å) and washed for 7 min with 0.1% TFA. The pre-column was subsequently connected to an analytical C18 column (ThermoFisher Scientific, Bremen, Germany, 75 μm × 50 cm, particle size 2 μm, pore size 100 Å). Peptide separation was performed with 400 nL/min flow rate with a gradient that was initiated with 95% A (0.1% formic acid) and 5% B (84% acetonitrile, 0.1% formic acid) with increasing amounts of B (up to 40% within 95 min). The concentration of B was then increased to 95% within 2 min and maintained for 3 min. Afterwards, the column was again adjusted to 5% B. The LC system was directly coupled with a nano electrospray ionization source (ThermoFisher Scientific, Bremen, Germany) to the Orbitrap Elite mass spectrometer. The system operated with a scan range from 300 to 2000 *m*/*z* with a resolution of 30,000 and 500 ms maximum acquisition time. From each full scan, the 20 most intensive ions were selected for low-energy collision-induced dissociation (CID) with 35% collision energy and 50 ms maximal acquisition time. After fragment ion (MS/MS) scans, the mass-to-charge (*m*/*z*) values of the precursor masses were maintained for 35 s on a dynamic exclusion list. The mass spectrometry raw data have been deposited to the ProteomeXchange Consortium [25] via the PRIDE partner repository with the data set identifier PXD068000.

### 2.5. Data Analysis

Resulting raw files were analyzed by MaxQuant (v.2.4.9.0), whereby recommended vendor settings were chosen with the exception that trypsin was specified as a digestion enzyme and oxidation was chosen as variable modification [26,27]. The human reference proteome FASTA (downloaded from https://www.uniprot.org/, accessed on 15 August 2022, 79,740 entries) was used to assign proteins [28].

The normalization and statistical analysis were performed using R version 4.4.0 (R Core Team, 2024) and the ProtStatsWF package [29]. For preprocessing of the raw protein group intensities from MaxQuant, median, quantile, and LOESS normalization methods were applied after log2-transformation and compared [30]. Based on boxplots, principal component analysis (PCA) plots, and MA-plots, LOESS was identified as the most suitable normalization method.

For a comparison of individual cell lines, we performed a one-way Welch ANOVA (accounting for possibly different variances between groups). The ANOVA was only performed if at least 3 valid values were present per group. The ANOVA *p*-value was adjusted according to the Benjamini–Hochberg procedure. Welch *t*-tests between pairs of groups were performed and adjusted with the Bonferroni–Holm procedure for each protein as a post hoc test.

A protein was considered significantly differentially expressed if the ANOVA *p*-value, after Benjamini–Hochberg FDR adjustment, was <0.05 in combination with a post hoc Welch *t*-test *p*-value < 0.05. A cut-off regarding the fold change (DMD/WT and WT/DMD) was set at ≥1.5.

The significantly expressed proteins were then analyzed regarding the respective localizations and functions of the proteins contained within them using the Gene Ontology features of DAVID. To accomplish this, the terms Cellular Compartment (CC), Biological Process (BP), and Molecular Functions (MFs) were used [31,32]. The threshold was set at *p*-value < 0.05. Heat maps and volcano plots were generated with GraphPad Prism Version 10.6.1. For the heat maps, the LFQ intensities were averaged and normalized with the Z-Score. Venn Diagrams were generated with Venny 2.1 [33].

## 3. Results

In the course of this project, a proteomics probe was prepared from human iPSC-derived skeletal muscle cells in a two-dimensional cell culture model and an organoid model. The usage of induced pluripotent stem cell (iPSC)-based cell culture models for skeletal muscle research offers the advantage of greater material availability and ease of access as expandable iPSC lines are more easily generated from blood cells or fibroblasts compared to acquiring muscle biopsies with limited expandability from patients.

We acquired three DMD iPSC lines, two from Children’s Hospital Boston [20] and one from UCSD [21], which were fully characterized. Concerning the DMD geno- and phenotypes, DMD-iPS1 and DMD-iPS2 (CHB, Boston, MA, USA) were induced in a six-year-old male patient with an identified deletion of exon 45-52 of the dystrophin gene and a clinical DMD phenotype, while DMD3 (UCSD061i-65-1) (UCSD, San Diego, CA, USA) was induced in a 23-year-old male patient with an undisclosed dystrophin mutation, clinical DMD, and a dilated cardiomyopathy phenotype. We did not attempt to create an isogenic line from DMD1 as a control and instead used iPSC we derived from cord blood [22] as the control group, which clearly constitutes a limitation of our study. Cells were differentiated according to the 2D protocol developed by Chal et al. (2015) [15] as well as our 3D organoid differentiation system reported in Mavrommatis et al. (2023) [16,18] toward skeletal muscle cells, from which extracts were taken and prepared for mass spectrometry.

Transcriptomic validation of myogenic identity in the two used differentiation systems was achieved through scRNA-seq analysis of canonical muscle as described by us before [19] (Appendix A). Presence of developmental myosin heavy chain isoforms MYH3 (embryonic) and MYH8 (neonatal) confirms the fetal-to-neonatal maturation trajectory. The structural marker DES (desmin) demonstrates muscle fiber integrity and PAX7, MYOD1, and MYOG distributions confirm the myogenic progenitor identity within clusters of myogenic progenitors.

### 3.1. General Mass Spectrometry Analysis

The utilization of mass spectrometry confers the benefit of the acquisition of extensive data sets with a minimal requirement for sample volumes. Using mass spectrometry, we were able to identify 3007 proteins in 2D cell culture samples and 2709 proteins in the 3D organoid cell culture samples. It was evident that for both culture methods, WT and DMD samples could be distinguished by their protein profile, as visualized by principal component analysis (PCA, Figure 1a,b).

Here, biological replicates of WT and DMD cell lines clustered in separate groups, whereby DMD1 and DMD2 clustered in closer proximity, while DMD3 formed a distinct separate group, indicating greater disparities in the proteome. To assess similarities between WT and DMD cell lines, quantified proteins were compared to each other, resulting in an overlap of 1685 proteins for 2D-cultivated cells and 1811 proteins for 3D-cultivated cells (Figure 1c,d). However, several proteins were found to be exclusively identified in one of the sample types (Appendix A). Based on GO Term Analysis of the overlapping proteins (Appendix A) we could define a core proteome of the cultured muscle samples. Notable examples include Myotilin, Myozenin-2, Myosin isoforms 2, 3, 4, 7 and 8, Tropomyosin-3 as well as Troponins C and T.

We further supported the high similarity between the different replicates by Pearson correlation, with correlation coefficients of r = 0.88 in WT 2D samples and DMD 2D correlation coefficients of DMD1 r = 0.846, DMD2 r = 0.865, and DMD3 r = 0.872. In WT 3D samples (r = 0.901) and DMD 3D (DMD1 r = 0.899, DMD2 r = 0.905, DMD3 r = 0.915), a high similarity between replicates was observed (Figure 1e). Comparison between proteins identified in at least one replicate of the 2D-cultivated and 3D-cultivated cells revealed a high overlap of 62.5% (2198 proteins) (Figure 1f). Nevertheless, cells grown in 2D displayed 809 and cells cultivated in 3D displayed 511 unique proteins (Figure 1f and Appendix A).

### 3.2. Analysis of Individual Proteins

To assess the influence of a mutated DMD gene on the overall proteome, a relative quantification was performed, identifying proteins significantly differentially expressed between wild-type and DMD cell lines. Prior to assessment, stringent filtering criteria were applied, whereby only proteins which had been quantified in a minimum of 70% of the replicates were considered for further analysis. This resulted in 1680 quantifiable proteins in the 3D samples and 1522 quantifiable proteins in the 2D samples (Appendix A). Subsequent statistical evaluation identified 338 proteins being significantly differentially expressed between the 3D samples. Among these, 49 proteins were significantly differentially expressed between all DMD muscle organoids and the wild-type organoids, with 20 proteins being of higher and 29 proteins of lower abundance in DMD organoids (ANOVA *p*-value < 0.05, Benjamini–Hochberg FDR *p*-value < 0.05, Post hoc *p*-values < 0.05, Appendix A). In the 2D samples, 83 proteins were found to be significantly differentially expressed (ANOVA *p*-value < 0.05, Benjamini–Hochberg FDR *p*-value < 0.05, Post hoc *p*-values < 0.05, Appendix A). Out of these, 51 showed higher abundance in 2D DMD samples, while 32 showed lower abundance. Notably, in the 2D samples, these changes in abundance were mainly observed in the DMD3 cell line compared to the wild type, whereas only one protein was differentially expressed across all three cell lines. In contrast, differential expression of proteins occurred more evenly across cell lines in the organoid samples (Appendix A).

In the first step, we aimed to annotate proteins which were identified as differentially expressed in all 3D DMD cell lines compared to their respective WT counterparts (Figure 2a,b) to potentially reveal common alterations caused by mutations in the DMD gene. To do so, gene ontology (GO) term enrichment analysis was conducted (Appendix A). With this, we were able to conclude that proteins found to be of higher abundance in DMD cell lines could be mainly annotated to the extracellular region (nine proteins, GO Term enrichment analysis: *p*-value 2.8 × 10^−4^), including Emilin-1, Collagen Type VI Alpha 2, Apolipoprotein E, Collagen Type XVIII Alpha 1 Chain, Annexin A6, Serpin H1, Nidogen 2, Collagen Type VI Alpha 3 Chain and Annexin A2, and to the sarcolemma (four proteins, GO Term enrichment analysis: *p*-value 2.8 × 10^−4^), including Collagen Type VI Alpha 2 Chain and 3 Chain, and Annexins A6 and A2 (Figure 2a). Functionally, upregulated proteins were mainly involved in actin cytoskeleton organization (GO Term enrichment analysis: *p*-value < 0.05), cell adhesion (GO Term enrichment analysis: *p*-value 0.001), and collagen fibril organization (GO Term enrichment analysis: *p*-value 0.002). Proteins displaying the highest fold changes among all DMD cells lines compared to the WT included, for example, Filamin-B, Collagens (COL6A2, COL6A3, COL18A1), and Myosin 9 (Table 1).

The 29 proteins identified as being less abundant in DMD organoids compared to the control samples were mainly located in and associated with mitochondria (GO Term enrichment analysis: *p*-value = 1.5 × 10^−2^) and the cytosol (GO Term enrichment analysis: *p*-value = 3.0 × 10^−6^) (Figure 2b). Functionally, these proteins mainly belonged to intracellular energy metabolism processes (glucose metabolic process GO Term enrichment analysis: *p*-value 3.9 × 10^−11^, glycolysis GO Term enrichment analysis: *p*-value 0.002, Appendix A).

Out of the 83 significant differentially expressed proteins in the 2D samples, 58 were found to be of higher abundance in DMD, primarily in the DMD3 cell line (Figure 3a). Regarding location, proteins were associated with the extracellular matrix (34 proteins, GO Term enrichment analysis: *p*-value 5.3 × 10^−18^) and the cytoskeleton (actin cytoskeleton, GO Term enrichment analysis: *p*-value 4.1 × 10^−4^, and Z disk, GO Term enrichment analysis: *p*-value 6.6 × 10^−2^). This is of particular interest given the established role of the Z disk in DMD, where it stabilizes the structure of the muscle cell and the arrangement of actin and myosin filaments. Dystrophin is a protein that, under normal circumstances, connects the Z disk to the muscle cell. Mutations in the DMD gene result in Z-disk instability, which consequently leads to muscle wasting. For this reason, a more detailed examination of these proteins was undertaken. Identified proteins associated with the Z disk and upregulation in DMD were as follows: Alpha-actinin-1, Alpha-actinin-4, Myosin regulatory light polypeptide 9, Myosin regulatory light chain 12B, and Myosin regulatory light chain 12A. Further, functionally, five proteins upregulated in DMD were associated with muscle contraction (GO Term enrichment analysis *p*-value 1.5 × 10^−2^), including Myosin light polypeptide 6, Tropomyosin alpha-1 chain, Myoferlin, Caldesmon, and Tropomyosin alpha-4 chain.

It should be further mentioned that Caldesmon, Myosin 9, Tropomyosin Alpha-1-Chain, and Collagen 18A were among the proteins displaying the highest fold change between DMD and WT samples from both cultivation systems (Table 1 and Table 2).

Concerning the 2D DMD samples, 25 proteins were found to be of lower abundance (Figure 3b). Here, GO Term analysis revealed that these proteins were mainly associated with the nuclear compartment (19 proteins, GO Term enrichment analysis: *p*-value 4.29 × 10^−4^, Appendix A). Functionally, the downregulated proteins were mainly involved in DNA recombination (GO Term enrichment analysis *p*-value 0.006), mRNA splicing (GO Term enrichment analysis *p*-value 0.007), and mRNA transport (GO Term enrichment analysis *p*-value < 0.01). Notably, RUVBL1 and BASP1 were found to be of lower abundance in both 2D and 3D DMD cells lines, potentially indicating common alterations, independent of the cultivation methods (Table 1 and Table 2). Heat maps demonstrate a differential expression of Myosin 9, Collagen 18A, Tropomyosin 1, NCAM1, BASP1, and RUVBL1 in the DMD vs. the WT samples, specifically protruding in the 3D culture system (Figure 3c,d).

## 4. Discussion

Duchenne muscular dystrophy is a fatal muscular disease caused by mutations in the DMD gene, in turn causing severe symptoms such as muscle weakness and wasting, ultimately leading to a shortened life span in affected individuals. Establishing suitable in vitro models is crucial to determine pathophysiological events on the molecular level as well as to investigate potential treatment options. Here, we pattern DMD patient-derived iPSCs toward skeletal muscle cells and provide one of the first DMD patient proteome data sets from DMD iPSC-derived muscle.

For that, we have established proteomics probe preparation being applicable to two-dimensional and three-dimensional cultures from human iPSC-derived skeletal muscle cells. The analysis was performed on samples after Day 35 of differentiation to skeletal muscle, when spontaneous myotube contractions indicated functional maturation. The availability of more material, which is easier to cultivate, expand and investigate, is an advantage of our iPSC-derived skeletal muscle samples compared to muscle biopsies in case of DMD. Proteomic analysis utilizing mass spectrometry offers the global identification of proteins and the determination of their relative abundance within different cell lines.

Our analysis identified 3007 proteins in 2D cultures and 2709 proteins in 3D organoids. Proteins associated with the extracellular matrix, cytoskeleton, and Z disk were found to be upregulated in DMD cells, consistent with known pathological hallmarks of DMD. In particular, collagens (COL6A2, COL6A3, COL18A1), Filamin-B, and Myosin isoforms displayed high fold changes, highlighting alterations in structural and contractile components (Table 1 and Table 2). Proteins involved in mitochondrial metabolism were observed to be downregulated, especially in DMD3, indicating potential defects in energy homeostasis. Notably, some dysregulations, such as the reduced abundance of RUVBL1 and BASP1, were common to both 2D and 3D cultures, suggesting cultivation-independent alterations.

Other research groups have mainly focused on material from muscle biopsies or animal models like the mdx mouse model [34,35]. Besides ours, only one study has provided proteomics data of iPSC-derived muscle tissue in DMD patterned in a two-dimensional culture system [36]. Mournetas et al., 2021, detected dysregulations in DMD in the early somite stage of muscle differentiation. Similar to our proteomic analysis, fibrosis-related proteins were identified as being upregulated in their analysis, i.e., collagens (COL1A2) [36]. Additionally, Reggio et al., 2025, recently performed a proteomic comparison of 3D skeletal muscle–pericyte aggregates and 2D cultures, demonstrating that the 3D system promotes the upregulation of muscle matrix and contractile proteins [37].

Our findings extend this knowledge by providing a comprehensive proteomic profile at a later stage of differentiation, including both 2D and 3D systems.

Interestingly, when comparing the proteome of 2D and 3D cells, a large overlap could be achieved, indicating that both cultivation methods lead to the expression of largely similar proteins (Figure 1f). This supports our previous assessment that the 3D organoid protocol does not go beyond the 2D protocols when it comes to provide maturated physiologically responsive skeletal muscle cells [17]. The main advantage of 3D organoids has been demonstrated in providing maturated skeletal muscle stem cells in the dormant, activated, and committed stages, which has been demonstrated with scRNA-seq analysis [19]. As the various subsets of cell types were not separated before proteomic analysis in our current study design, we are not expecting that the “bulk-proteomics” approach used in the current study can disclose alterations in the DMD muscle stem cell pools versus the control group. The established 2D and 3D differentiation protocols were utilized as two independent sources of DMD-specific proteins for comparative analysis.

Using this approach, we further identified 49 proteins in the 3D samples and 83 proteins in the 2D samples to be differentially expressed in Duchenne muscular dystrophy (DMD) patient iPSC-derived skeletal muscle cells compared to human wild-type iPSC-derived skeletal muscle cultures.

Regarding functional groups and localization, proteins found to be of higher abundance in DMD cell lines were primarily annotated to the extracellular matrix and cytoskeleton being involved in muscular contraction. Downregulated proteins in DMD were mainly localized in the nucleus or mitochondria, with functional roles in transcription, translation, and energy metabolism. Given that DMD is a disorder affecting the actin cytoskeleton, influencing muscle contraction, it was expected that differentially expressed proteins were associated with muscle development, function, and pathophysiological mechanisms in DMD (Table 1 and Table 2).

Thus, in the following, we are discussing the six proteins, which have been significantly differently expressed in the 2D as well as the 3D system in DMD lines (Figure 3c,d), concerning their reported or potential role in DMD pathophysiology:

MYH9 (Myosin 9), a protein of the Myosin family, is a cytoskeletal protein responsible for cell shape and intracellular transport. Overall, we identified an increased abundance of cytoskeletal proteins in DMD samples. A significant upregulation of MYH9 was detected in DMD samples of both 2D and 3D samples (2D DMD3 Tukey-test *p*-value < 0.05, 3D Tukey-test *p*-values < 0.005). Alterations in expression and increased oligomerization could also be verified in the *mdx* mouse model [38]. Compensatory processes, e.g., repair mechanisms of the cytoskeleton and sarcolemma, are one potential explanation for the higher abundance of MYH9, since mutations in the DMD gene are known to cause detrimental defects in muscle structure and function [38].

COL18A (Collagen 18A), as a protein of the extracellular matrix (ECM), is significantly overexpressed in DMD probes (2D DMD3 Tukey-test *p*-value < 0.05, 3D Tukey-test *p*-values < 0.05). Other collagen-related genes overexpressed in the 3D models are, for example, Collagen 6A2 and 6A3. Different collagens are expressed as core parts of the ECM in skeletal muscle. Collagens contribute to the ECM structure in skeletal muscle, and their accumulation is a hallmark of fibrosis in DMD [39].

TPM1 (Tropomyosin alpha 1-chain) belongs to the contractile apparatus of skeletal muscle cells. A significantly higher abundance was detected in samples of the DMD3 cell line (2D Tukey-test *p*-value < 0.05, 3D Tukey-test *p*-value < 0.05). Tropomyosin expression has been found to be age-dependent in proteomic analysis of the *mdx* mouse [38]. An upregulation of Tropomyosin as a compensation mechanism in the contractile apparatus while Dystrophin is lacking can be seen in the early stages of the diseases until fibrosis, myonecrosis, and replacement with fatty tissue take place.

NCAM1 (neural cell adhesion molecule or CD56) plays a role in myogenic differentiation. The protein is involved in cell-to-cell interaction and is a surface marker, where it has been described as a marker of satellite cells committed to myogenic differentiation [40]. In this regard, CD56 has been used in different protocols for the isolation of muscle stem cells [41]. A significant downregulation of NCAM1 in DMD samples can be observed in our analysis (3D Tukey-test *p*-values < 0.05, Table 1), possibly indicating a reduced regenerative capacity in DMD. This is in line with previous findings showing that a lack of Dystrophin may lead to less satellite cells committing to differentiation [6]. Therefore, our models, mainly the 3D models, could furthermore be an example for generating adult muscle cells. Using CD56 as a surface marker for satellite cells, further investigations based on the protocols could be performed.

BASP1 (Brain acid soluble protein 1) is a membrane-located protein and is represented in axonal neurons. BASP1 has an inhibitory function to MYC and calmodulin [42]. In our probes of the DMD3 cell line, we found significantly lower levels of BASP1 (2D Tukey-test *p*-value < 0.05, 3D Tukey-test *p*-value < 0.05). Since calcium-dependent processes are critical in DMD pathophysiology, reduced BASP1 may contribute to calcium-related dysfunction [43].

RUVBL1 (RuvB-like 1), being involved in transcriptional processes as a nuclear protein, has so far not been described in the context of DMD. The RUVBL1/2 complex has a role in pro-inflammatory processes toward macrophages [44]. We have found RUVBL1 to be significantly downregulated in DMD3 samples (2D Tukey-test *p*-value < 0.05, 3D Tukey-test *p*-value < 0.05). Therefore, limitations of the cellular 2D and 3D models might lie in the pathophysiological processes of inflammation, as various cells of the immune system are not involved in the patient-derived iPSC skeletal muscle model.

We consider these proteins from various cellular compartments discussed above as potential candidates for further investigation and as possible biomarkers in DMD. Furthermore, these proteins are involved in different processes of DMD pathophysiology (calcium dysregulation, reactive oxygen species, inflammation, expansion of ECM, and fibrotic processes).

As a general conclusion from using the two-dimensional as well as the organoid culture systems for initiating differentiation from the same DMD iPSC lines, results from the analysis of the two-dimensional culture system show greater variability between the different DMD cell lines than the results from the analysis of skeletal muscle organoids, possibly indicating an advantage of the organoid over the two-dimensional culture system in providing skeletal muscle cells in vitro for disease modeling.

Three-dimensional muscle models, including our organoid system, may serve as valuable platforms for proteomic analyses, particularly as they have the potential to better recapitulate the muscle environment, extracellular matrix, and mature muscle fibers [45]. Two-dimensional iPSC-based systems, such as those described by Chal et al., 2015, can effectively mimic key aspects of myogenesis at the molecular level [15]. However, limitations regarding structural and functional maturation remain inherent to 2D systems, which has motivated the development of various 3D approaches employing distinct strategies to overcome these shortcomings [45]. Accordingly, different 3D approaches may provide complementary insights from a proteomic perspective, depending, for example, on the use of specific scaffolds [46,47] or on co-culture strategies with additional cell types such as adipocytes, fibroblasts, or pericytes [37,48,49].

In this context, we must acknowledge the limitations of our overall study design concerning the statistical significance of the conclusions stated above. We have presented data from two differentiations (2D/3D) for each of the three DMD lines versus two differentiations (2D/3D) for the control group. DMD1 and DMD2 are iPSC lines differentiated from the same patient. This experimental design has contributed to the closer clustering of DMD1- and DMD2-derived data versus the DMD3 group. Further provision of proteomic data sets derived from further DMD patient iPSC lines will be necessary to confirm our panel of differentially expressed proteins.

Developments in proteomic methodology toward “single-cell proteomics” might also allow us to specify skeletal muscle stem cell subpopulations and their pathophysiological alterations as currently already established on the transcriptomics level with scRNA-seq [19,50,51].

## 5. Conclusions

In this study, we performed 2D and 3D differentiation of DMD patient-derived induced pluripotent stem cells with the aim of providing proteome data sets. Proteomic profiling of human DMD samples may help identify new biomarkers and signaling pathways that could serve as potential therapeutic targets. Our findings show similarities to previously reported proteomic analyses of mdx mouse models, particularly highlighting differential expression in the cytoskeleton and extracellular matrix compartments. Based on our—concerning sample size restricted—data sets, we propose MYH9, COL18A, TPM1, NCAM1, BASP1, and RUVBL1 as proteins specifically altered in DMD, representing promising candidates for further validation in additional studies.

## Figures and Tables

**Figure 1 cells-14-01688-f001:**
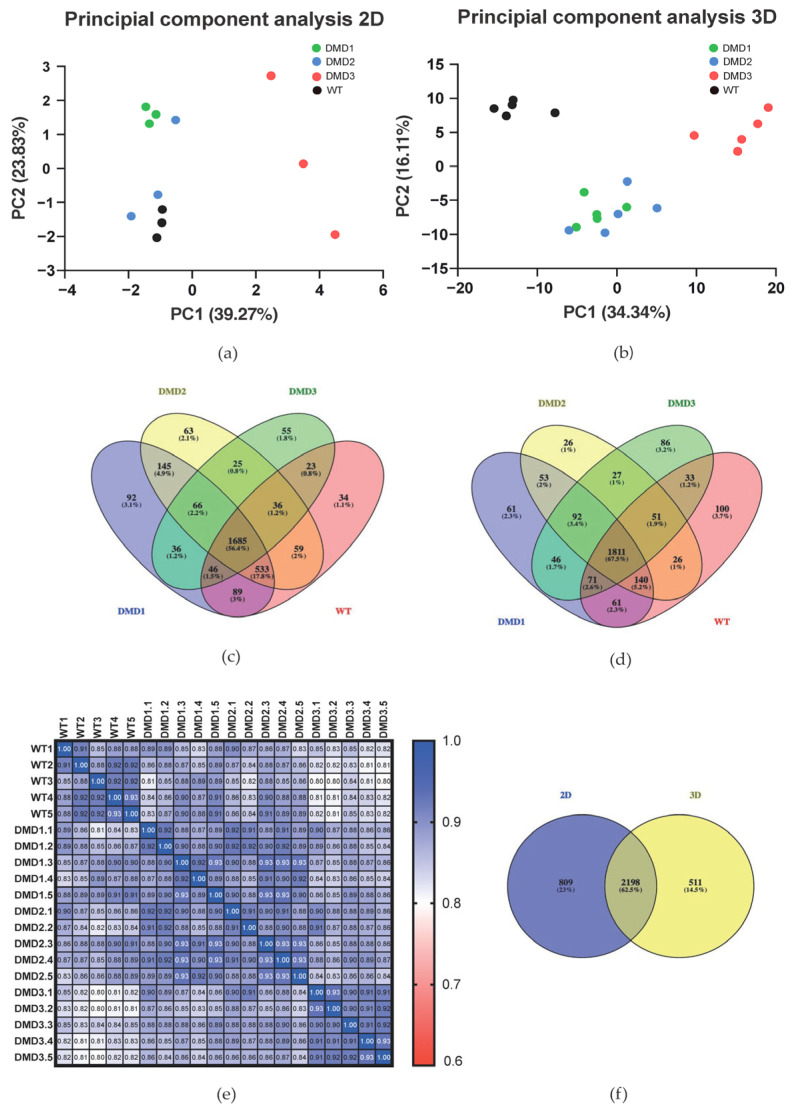
(**a**,**b**) Principal component analysis. Legends on the right. (**a**) PCA based on 2D samples presenting clustering of biological replicates of each cell line with samples of DMD3 clustering apart. (**b**) PCA based on 3D organoid samples showing clustering of the biological replicates from each tested cell line. (**c**,**d**) Venn diagrams showing the overlap between the analyzed cell lines. (**c**) 2D Venn diagram showing the overlap between WT, DMD1, DMD2, and DMD3 in the 2D samples with an overall overlap of 56.4%, showing a valid reliability between the different cell lines. (**d**) Venn diagram of the 3D samples representing an overall overlap of 67.5%. Diagrams generated with Venny 2.1. (**e**) Pearson correlation exemplarily between the 3D replicates. Correlogram based on the Pearson correlations of the 3D samples. High correlation between the replicates can be detected. Every cell annotated with the r (Pearson correlation). Figure created with GraphPad Prism. (**f**) Venn diagram comparing detected proteins in 2D and 3D samples, showing an overlap of 2198 proteins (62.5%). Diagram generated with Venny 2.1.

**Figure 2 cells-14-01688-f002:**
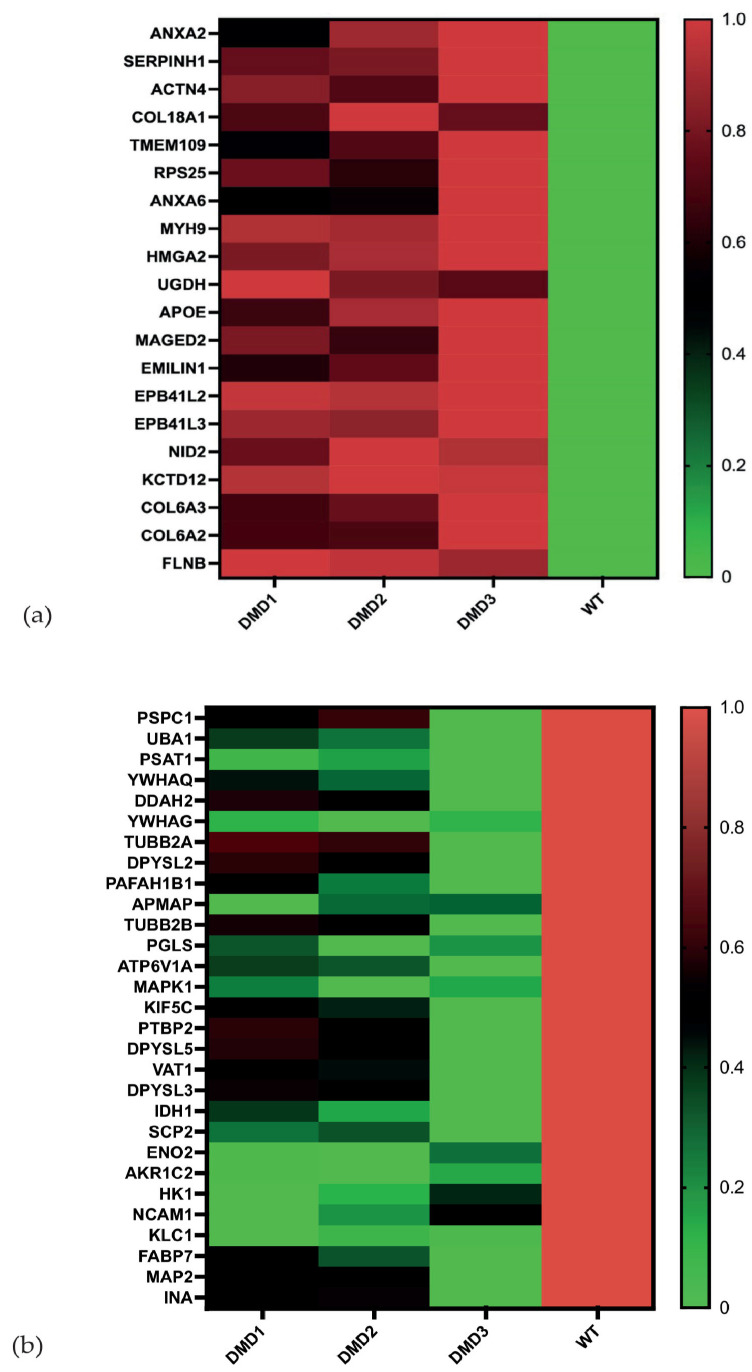
(**a**) Heat map of proteins significantly upregulated in DMD1, DMD2, DMD3 vs. WT in the 3D organoid. Based on the Z-Score, the heat maps were generated, with the colored legend on the right side. (**b**) Heat map of proteins significantly downregulated in DMD1, DMD2, DMD3 vs. WT in the 3D organoid.

**Figure 3 cells-14-01688-f003:**
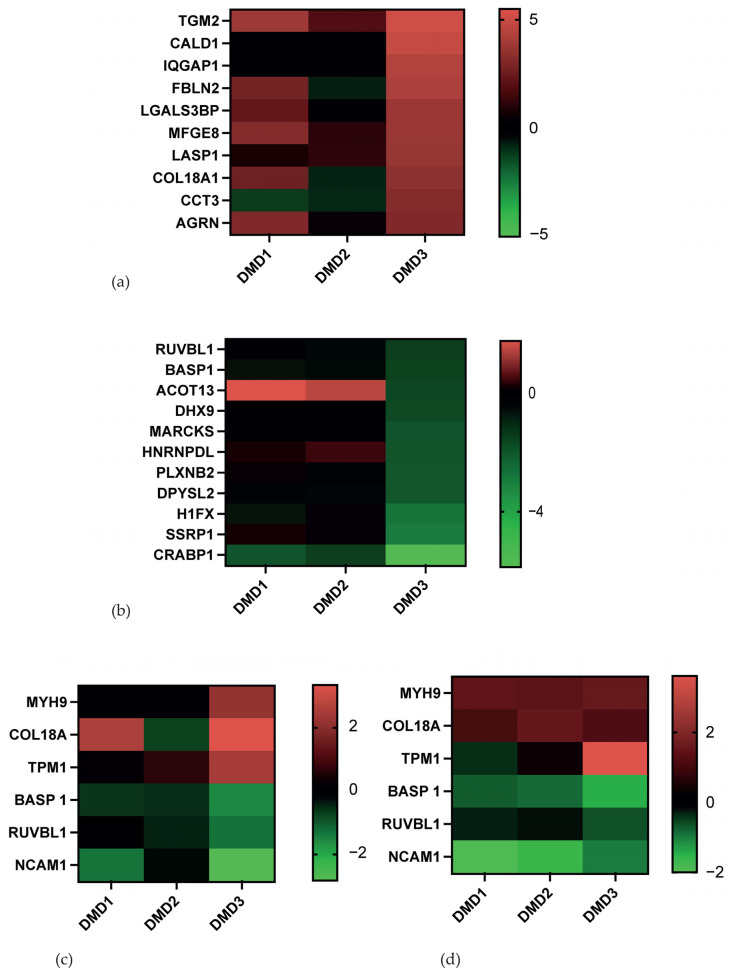
(**a**) Heat map of the top 11 proteins significantly upregulated in DMD1, DMD2, DMD3 vs. WT in the 2D samples. (**b**) Heat map of the top 11 proteins significantly downregulated in DMD1, DMD2, DMD3 vs. WT in the 2D samples (**c**,**d**) Heat map of selected proteins significantly expressed both in 2D and 3D; (**c**) 2D significant proteins; (**d**) 3D significant proteins. Legend: Differential expression shown as fold change relative to WT.

**Table 1 cells-14-01688-t001:** Selected proteins significantly expressed in 3D, * <0.05, ** <0.005, *** <0.0001, / non-significant.

Protein	Tukey-Test *p*-Value DMD1	Tukey-Test *p*-Value DMD2	Tukey-Test *p*-Value DMD3	ANOVA FDR *p*-Value	FC (DMD1/WT)	FC (DMD2/WT)	FC (DMD3/WT)
MYH9	0.0001 **	0.00009 ***	0.0002 **	0.00004 ***	2.67	2.59	2.88
COL18A	0.002 **	0.003 **	0.01 *	0.013 *	2.02	2.80	2.20
TPM1	/	/	0.014 *	0.016 *	0.77	1.24	12.21
BASP1	/	/	0.02 *	0.017 *	0.60	0.56	0.36
RUVBL1	/	/	0.018 *	0.022 *	0.82	0.87	0.64
NCAM1	0.009 *	0.004 **	0.048 *	0.0046 **	0.27	0.32	0.50

**Table 2 cells-14-01688-t002:** Selected proteins significantly expressed in 2D, * <0.05, ** <0.005, / non-significant.

Protein	Tukey-Test *p*-Value DMD1	Tukey-Test *p*-Value DMD2	Tukey-Test *p*-Value DMD3	ANOVA FDR *p*-Value	FC (DMD1/WT)	FC (DMD2/WT)	FC (DMD3/WT)
MYH9	/	/	0.008 *	0.007 *	1.01	1.11	4.31
COL18A	/	/	0.01 *	0.011 *	5.83	0.60	10.17
TPM1	/	/	0.002 **	0.0005 **	1.15	1.51	5.35
BASP1	/	/	0.005 *	0.006 *	0.66	0.71	0.34
RUVBL1	/	/	0.008 *	0.0006 **	1.03	0.76	0.41
NCAM1	/	/	/	/	0.38	0.80	/

## Data Availability

All data are available via ProteomeXchange [25] (https://www.proteomexchange.org) with the identifier PXD0068000.

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
