# Peer review of "Proteomics of Duchenne Muscular Dystrophy Patient iPSC-Derived Skeletal Muscle Cells Reveal Differential Expression of Cytoskeletal and Extracellular Matrix Proteins"

_cells, 2025, doi:10.3390/cells14211688_

Round 1
Reviewer 1 Report
Comments and Suggestions for Authors
The work by Gallert and colleagues presents a proteomic analysis on DMD patients’ derived iPSC cells, used as a viable alternative to studies conducted on patients’ muscle biopsies, which are of course very difficult to obtain.
I think the MS is important as it shows it is possible to employ iPSCs to conduct proteomics studies of DMD, and the results they report, specifically those regarding the identification of differentially regulated proteins of the ECM and cytoskeleton compartments (expected in a pathology such as DMD), in part confirm the validity of this approach.
There are although some limitations intrinsic to the analysis that need to be addressed.
Please find below a series of minor and major points that I am indicating in the order I have found them in the MS, from start to finish.
First of all, I reckon that the identified proteins should not be mentioned (i.e. named one by one) in the title of the MS – reference to ECM and cytoskeletal proteins, for example, should suffice I suppose? In any case I would reformulate the title in a more concise, or at least less specific, way.
Lines 64-66: typos/words redundancy
Line 78: missing word(s)
Line 80: reference jump from 18 to 42?
Duplication of prepositions, conjunctions, etc. (of of, to to, and and, and so on) along different sections of the MS, please check.
Numbering of the supplementary tables – not right order in the text (you cannot start from S7, and it should be easy enough to put it as S, I would think?), and also I cannot reconcile what you write in the Results with what the tables actually report – PLEASE CHECK if there is a problem in consistency, and make everything clearer. For example, TableS1 and Table S2 depict what in text is defined Table S4 and S3, etc.
Fig. 3 is very confusing! In both panel a and b data do not seem to be calculated as a comparison with the wt, but rather with DMD3? Accordingly, panels c and d are even more confusing… Unless I am missing something quite obvious, in which case I apologise, this figure should be re-made since, as it is, it does not provide useful information – quite the opposite, I’m afraid.
Line 375: please redefine the idea of model – animal cells are not organisms…
Lines 400-402: not really sure what this sentence mean?
Line 425: Please add reference to the mdx mice results mentioned
No detail is provided on the origin of the 3 different DMD cell lines, nor a discussion is made on the possible differences between the patients they derive from – age, sex, etc. This is a crucial point as it constitute the basis for understanding what is being analysed and what kind of significance can any result really have.
I had to go and find the original papers describing the origin of the iPSCs DMD lines, and discovered that DMD1 and 2 come from the same, 6YO patient (!), which in itself should be mentioned in the MS without the reader having to go and check. The origin of DMD3, on the other hand, is very hard to get from the publication cited (I gave up after a few minutes I was trying to identify the DMD patients within the paper). The fact that the first two cell lines are differentiated from the same patient should be clearly specified, as it might at least partially explain why their identified up- or down-regulated proteins cluster together in all of the analysis here presented – and this should be included in the discussion. Although I understand that employing both DMD1 and 2 might make the statistics better, the fact they have the same origin should be commented upon and not ignored, especially if then the DMD3 sample presents outliers. Which also takes me to the last important consideration below.
Lines 460-466: Not sure the conclusions drawn on all the proteins identified only in DMD3 have much value, as they come from one single cell line. It’s hard to see statistical significance in these results – any comment on these ‘uniquely identified’ proteins are highly speculative, and the Authors should state so, to the least. A discussion on this specific problem (DMD3 having more outliers than the other two lines) should be included to start with, and the lack of significance commented upon. As mentioned above, the origin of DMD3 is not easily found within, or inferred from, the citation given, which is a problem per se: please insert specifics of all the missing data on the DMD iPSC lines in the Mat and met section, and discuss it in the manuscript accordingly.
All in all the conclusions section is more cautious in indicating the real value of the analysis reported, without over-stating like the most speculative part of the discussion does (thus needing re-elaboration) without adequate support from robust statistical evidence.
Lines 484-487486: Not clear, please rephrase
Reviewer 2 Report
Comments and Suggestions for Authors
The present work reports the potential use of iPSCs derived from DMD individual to predict changes in protein abundance that are linked to DMD pathogenesis. While the overall concept sounds good the strategy adopted by authors has serious flaws limiting the interest of this referee. The DMD groups used is too low and the intrinsic sample to sample variability raised concerns about the reliability of the produced data. Moreover, some crucial details are not reported hindering the ability of this referee in judging the overall conclusions. Referee’s comment are appended below and must be followed to secure publication in this journal
Proteomics of skeletal muscular dystrophies is limited by the amount of protein which can be provided from patient biopsies. The sentence contains some imprecisions; please rephrase as it follows: Proteomics of dystrophic muscle samples is limited by the amount of protein that can be extracted from patient biopsies.
The percentage of myogenic differentiation of the different models used in this study is not reported, neither in 2d or 3d cultures conditions. This major limitation raised serious concerns about the overall study. Please provide information about the differentiation and fusion index of differentied DMD lines in comparison to the reference control both in 2d and in 3d cultures.
It is unclear to this referee the system used to generate 3d muscle organoids. Please describe in details the model and provide data demonstrating the need and the benefit for adopting the system. Recently, proteomics has been flanked to a 3D muscle surrogate generated with a novel 3D RoWs fabrication system. The authors are recommended to follow the workflow of this manuscript to better propose their own data.
“In comparison to biopsy samples, the usage of induced pluripotent stem cell (iPSC)-based cell culture models for skeletal muscle research offers the advantage of greater material availability and ease of access”. The author’s statement is not formally true since they are not providing comparative data that allow to compare differences protein abundance between iPSC-derived muscle cells and muscle biopsies. Please remove this part otherwise please provide supportive data.
PCA in figure 1A. A clear-cut separation between control samples and DMD1 is not appreciated. Such aspect is not discussed. The fact that these two proteome profiles are too similar affects potency of statistical test and also data reliability in the DMD group due to the high-sample to sample variability. The number of samples used for the study is too low. The data extracted of the authors have a limited reliability if tested on a large cohort of samples.
The authors are not providing details on the mutation harbored by the DMD background of iPSCs models. Control cells only a single control line is used. The findings of this manuscript fall short at convincing this referee.
It’s unclear for this referee if the pool of those that authors indicate as significantly modulated proteins is a pool of protein from three different statistical analysis (ANOVA p-value < 0.05, Benjamini Hochberg FDR p-value < 0.05, Post-hoc p-values < 0.05). please specify
The authors must discuss better their findings. How are the authors’s findings positioned when compared with established literature data? What is novel and also what is meriting attention in this manuscript.
Round 2
Reviewer 1 Report
Comments and Suggestions for Authors
The Authors have taken onboard the comments and concerns I expressed and addressed them in a satisfactory manner within the revised MS, which in this improved form I believe being suitable for publication
Reviewer 2 Report
Comments and Suggestions for Authors
In this revised version, authors clarified all the concerns of this referee by providing comments, text adjustment and additional experiments. Despite this, the small sample size tested remains as the major limitation, precluding broad generalization of authors' findings. However, the authors clearly reported this in their text as a study limitation. Therefore, in this form the manuscript can be considered suitable for this journal